

# Purification of plant-derived anti-virus mAb through optimized pH conditions for coupling between protein A and epoxy-activated beads

Ilchan Song[1],[*], Yang Joo Kang[1],[*], Su-Lim Choi[2], Dalmuri Han[3], Deuk-Su Kim[1], Hae Kyung Lee[3], Joon-Chul Lee[4], Jeanho Park[4], Do-Sun Kim[5] and Kisung Ko[1]

[1] Department of Medicine, College of Medicine, Chung-Ang University, Seoul, South Korea
[2] Protein Purification Laboratory, Biotech R&D Center, Amicogen, Jinju, South Korea
[3] Division of Bacterial Disease Research, Korea Centers for Disease Control and Prevention, Osong, South Korea
[4] Research Institute of Industrial Technology Convergence, Korea Institute of Industrial Technology, Ansan, South Korea
[5] Vegetable Research Division, National Institute of Horticultural and Herbal Science, Rural Development Administration, Wanju-gun, South Korea
[*] These authors contributed equally to this work.

Corresponding author
Kisung Ko, ksko@cau.ac.kr

## ABSTRACT

The main goal of this research was to determine optimum pH conditions for coupling between protein A and epoxy-activated Sepharose beads for purification of monoclonal antibodies (mAbs) expressed in plants. To confirm the effect of pH conditions on purification efficacy, epoxy-activated agarose beads were coupled to protein A under the pH conditions of 8.5, 9.5, 10.5, and 11.5 (8.5R, 9.5R, 10.5R, and 11.5R, respectively). A total of 300 g of fresh leaf tissue of transgenic *Arabidopsis* expressing human anti-rabies mAb (mAb^P) SO57 were harvested to isolate the total soluble protein (TSP). An equal amount of TSP solution was applied to five resin groups including commercial protein A resin (GR) as a positive control. The modified 8.5R, 9.5R, 10.5R, and 11.5R showed delayed elution timing compared to the GR control resin. Nano-drop analysis showed that the total amount of purified mAb^P SO57 mAbs from 60 g of fresh leaf mass were not significantly different among 8.5R (400 μg), 9.5R (360 μg), 10.5R (380 μg), and GR (350 μg). The 11.5R (25 μg) had the least mAb^P SO57. SDS–PAGE analysis showed that the purity of mAb^P SO57 was not significantly different among the five groups. Rapid fluorescent focus inhibition tests revealed that virus-neutralizing efficacies of purified mAb^P SO57 from all the five different resins including the positive control resin were similar. Taken together, both pH 8.5 and 10.5 coupling conditions with high recovery rate should be optimized for purification of mAb^P SO57 from transgenic *Arabidopsis* plant, which will eventually reduce down-stream cost required for mAb production using the plant system.

# INTRODUCTION

Plants have enormous potential as bioreactors for the large-scale production of therapeutic reagents such as recombinant vaccines and antibodies (*Kang et al., 2017*; *Kim et al., 2016*; *Lim et al., 2014*). One of the major factors limiting commercial advances in plant-derived pharmaceuticals is the high cost and inefficiency of purification (*Hussack et al., 2010*; *Tschofen et al., 2016*). Plant leaf extracts contain an array of indigenous proteins and other water-soluble cell components such as DNA, chlorophyll and other pigments, alkaloids, phenolics, polysaccharides, and proteases (*Wilken & Nikolov, 2012*; *Zhang et al., 2015*). These components can affect the quality and yield of final purified protein or reduce purification efficiency because of resin or membrane fouling (*Jha et al., 2016*; *Moussavou et al., 2015*). Thus, extraction conditions should be optimized through screening and evaluation of the tissue disruption technique, particle size distribution, buffer composition, plant tissue-to-buffer ratio, and subcellular compartment expression (*Park et al., 2015*; *Romanik et al., 2007*).

Staphylococcal protein A is one of the first discovered immunoglobulin-binding molecules and has been extensively studied during the past decades. Because of its affinity to immunoglobulins, protein A has been widely used as a tool in the detection and purification of antibodies, and its role has been further developed in one of the most commonly employed affinity purification systems (*Hober, Nord & Linhult, 2007*; *Liu et al., 2010*; *Mazzer et al., 2015*). However, protein A should be properly coupled to a matrix such as agarose beads, as the orientation of protein A when coupled to the bead affects the holding capacity of the mAb and hence can affect purification efficiency (*Groher & Suess, 2016*; *Welch et al., 2017*; *Zhang, Duan & Zeng, 2017*). Sulfhydryl groups are the most highly reactive nucleophiles with epoxides, requiring a buffered system in the range of pH 7.5–8.5 for efficient coupling (*Hermanson, 2013*; *Tehrani Najafian et al., 2017*). Amine nucleophiles react at moderate alkaline pH values, typically needing buffer environments of at least pH 9.0. The reaction of the epoxide functionalities with hydroxyls requires high pH conditions, usually in the range of pH 11–12. In this study, four different pH conditions (pH 8.5, 9.5, 10.5, and 11.5) were applied to manufacture epoxy-activated beads coupled to protein A. Research to develop protein A affinity chromatography for efficient purification of biotherapeutic proteins from plant biomass as a bioreactor has not yet been undertaken. The main goal of the present research was to determine coupling pH conditions between protein A and epoxy-activated agarose beads that could efficiently purify anti-rabies monoclonal antibodies (mAbs) derived from transgenic *Arabidopsis* plants. The purified plant-derived mAbs (mAb$^P$s) from four different resins and a commercially available protein A agarose resin as a positive control were compared for purification efficiency, purity, and neutralizing activity.

# MATERIAL AND METHODS

## Floral dip transformation

Plant expression vector pBI mAb 57 carrying anti-rabies virus mAb light chain (LC) and heavy chain (HC) fused to KDEL ER retention signal was transferred into *Agrobacterium*

*tumefaciens* strain GV3101::pMP90 by electroporation (Fig. 1A). *A. tumefaciens* carrying mAb[P]SO57 expression cassettes was cultured at 28–30 °C in LB with kanamycin for 2 days. Agrobacteria were centrifuged (4,000 rpm, 10 min), and the pellets were resuspended with infiltration media (4.3 g/L of MS salts, 30 g/L of sucrose, 0.1 g/L of myo-inositol). Wild type Col-0 *Arabidopsis* plants were transformed using the floral dip method (*Clough & Bent, 1998*). Col-0 *Arabidopsis* seeds were sown on soil, and seedlings were grown for 4 weeks for floral dip transformation under standard conditions (16 h light/8 h dark cycle, 22 °C) in a growth chamber. To induce proliferation of many secondary bolts, the first bolts of *Arabidopsis* were trimmed off. The plant pots were inverted into the infiltration solution containing 0.02% Silwet L-77 for 5 min at a time, and the infiltrated plants were covered with black plastic bags for 2 days to ensure high humidity. On the following day, the plastic bags were removed, and the plants were maintained in the standard conditions in a growth chamber until seeds were well ripened. Then, the obtained seeds were sown on agar plates containing Murashige and Skoog (MS) medium (pH 5.7) (10 g/L of sucrose, 8 g/L of plant agar, and 4.3 g/L of MS B5 vitamin (Duchefa Biochemie, Haarlem, Netherlands)), containing 50 mg/L kanamycin and 25 mg/L cefotaxime to select true leaf generating shoots. For a further study, all shoots with green leaves were transplanted to a soil pot and maintained in a growth chamber at 22 °C under a 16 h light/8 h dark cycle.

## Genomic DNA preparation and PCR analysis

Plant genomic DNA preparation and polymerase chain reaction (PCR) analysis were conducted as described in *Song et al. (2015)*. Genomic DNA was isolated from the rosette leaves of *Arabidopsis* plants using a DNA extraction kit (RBC Bioscience, Seoul, Korea), according to the manufacturer's recommendations. PCR analysis was performed to confirm the presence of HC (281 bp) and LC (227 bp) genes of $T_1$ plants. Primer sets were described *as follows:* HC forward primer, 5′-CAG ACT CAC CAT TAC CGC-3′; HC reverse primer, 5′-AGT AGT CCT TGA CCA GGC-3′; LC forward primer, 5′-CAC TGG AAC CAG CAG TGA-3′; LC reverse primer, 5′-TGT AGT CGC CTG CAT ATG A-3′. The PCR reaction was subjected to 26 cycles of 94 °C for 20 s, 58 °C for 10 s, and 72 °C for 30 s. The PCR products were analyzed by electrophoresis in a 1.0% agarose gel with ethidium bromide, and visualized under UV illumination. The pBI mAb 57 vector was used as a positive control, and genomic DNA extracted from Col-0 *Arabidopsis* was used as a negative control.

## Western blot analysis

Western blot analysis was conducted as previously described in *Lu et al. (2012)*. To confirm protein expression levels of transgenic *Arabidopsis*, 80 mg of fresh leaf tissue, which had previously been confirmed as having the target mAb[P]SO57 protein genes (mAb SO57 HC (281 bp), LC (227 bp)) by PCR, was ground in 300 μL of $1\times$ PBS buffer (137 mM NaCl, 2.7 mM KCl, 10 mM $Na_2HPO_4$, 2 mM $KH_2PO_4$, pH 7.4). Homogenates were boiled with sample buffer (1M Tris-HCl, 50% glycerol, 10% SDS, 5% 2-mercaptoethanol, 0.1% bromophenol blue) and cooled in ice. After centrifugation for 5 min at 10,200×*g*,

A)

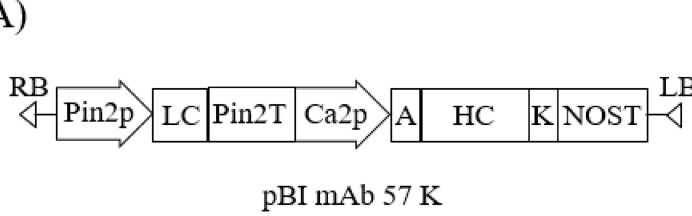

pBI mAb 57 K

B)

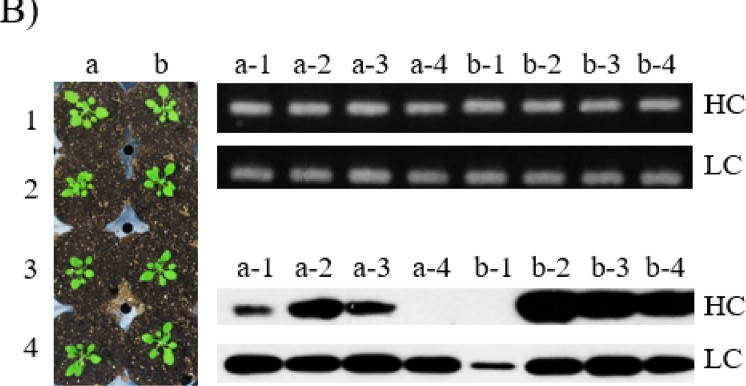

C)

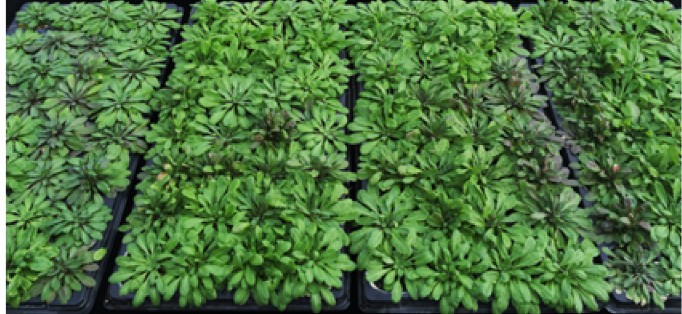

**Figure 1 Generation of transgenic *Arabidopsis* to express anti-rabies virus mAb$^P$SO57.** (A) Schematic diagram of vector constructs for expressing anti-rabies virus mAb$^P$SO57. Pin2p, promoter of Pin2 gene from potato; and Ca2p, cauliflower mosaic virus 35S promoter; control expression of light and heavy chains, respectively. K, KDEL, the 3′ endoplasmic reticulum (ER) retention motif; A, an alfalfa mosaic virus untranslated leader sequence of RNA4; Pin2T, terminator of Pin2 gene from potato; NOST, terminator of nopaline synthase (NOS) gene. (B) Selection and screening of $T_1$ transformants. Shoots that survived under kanamycin antibiotic selection were transferred to a soil pot and placed in a growth chamber under 16 h light/8 h dark cycle at 23 °C (left). PCR and western blot analyses of $T_1$ generation plants to confirm heavy chain (HC) and light chain (LC) gene existence and expression level. The genomic DNA fragments were extracted from fresh leaf tissue, amplified, and separated on a 1% agarose gel using electrophoresis. HC (50 kDa) and LC (25 kDa) were detected with HRP-conjugated goat anti-human IgG Fc- or IgG F(ab′)$_2$-specific antibodies, respectively. Lane 1–8, $T_1$ transformants putatively expressing anti-rabies mAb$^P$SO57. (C) Biomass production and purification of transgenic *Arabidopsis* expressing mAb$^P$SO57. Seeds were sown in a greenhouse, under conditions of 24 °C, 30% humidity, and 16 h light/8 h dark cycle.

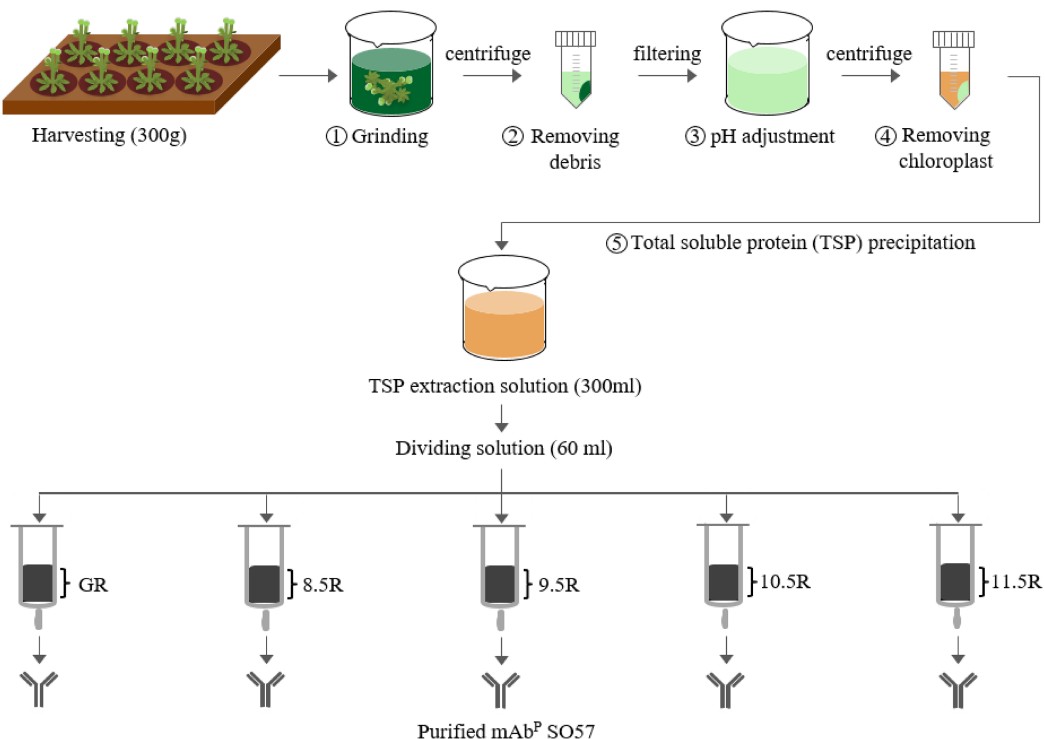

**Figure 2 Purification of anti-rabies mAb^P^SO57 with different resins.** Total soluble protein (TSP) extraction and purification process to purify mAb^P^SO57 with four differently coupled resins, and a control resin. The purification process was modified from *Park et al. (2015)*. Epoxy-activated agarose beads were coupled to protein A under the pH conditions of 8.5, 9.5, 10.5, and 11.5 (8.5R, 9.5R, 10.5R, and 11.5R, respectively). Commercial protein A resin (GR) (GE Healthcare, Uppsala, Sweden) was used as a positive control.

20 µL of supernatant was loaded in 12.5% SDS–PAGE gel and transferred to a nitrocellulose membrane (Millipore, Billerica, MA, USA). The membrane was then blocked with 5% skim milk (Sigma, St. Louis, MO, USA) in 1× PBS-T (1× PBS plus 0.1% (v/v) Tween 20) for 2 h at 25 °C. After blocking treatment, the blots were incubated with peroxidase-conjugated goat anti-human IgG Fcγ (Product # 109-035-008; Jackson Immuno Research Labs, West Grove, PA, USA) and anti-human IgG F(ab′)$_2$ (Product # 109-036-006; Jackson Immuno Research Labs, West Grove, PA, USA), which recognize the HC and LC of mAb^P^SO57, respectively. Protein bands were detected with the SuperSignal West Pico Chemiluminescent Substrate (Thermo Scientific, Rockford, IL, USA) and visualized using X-ray films (Fuji, Tokyo, Japan). The leaf of Col-0 *Arabidopsis* was used as a negative control.

## Purification of mAb^P^SO57 from plant leaf using different resins

For the purification of the plant-derived anti-rabies mAb^P^SO57, 350 g of *Arabidopsis* leaf tissue was homogenized in a HR2094 blender (Philips, Seoul, Korea) using extraction buffer (37.5 mM Tris-HCl, pH 7.5; 50 mM NaCl; 15 mM EDTA; 75 mM sodium citrate; 0.2% sodium thiosulfate) as previously described in *Park et al. (2015)* (Fig. 2). Ground samples were centrifuged at 9,000×*g* for 30 min at 4 °C, then the supernatant was filtered through a Miracloth (Biosciences, La Jolla, CA, USA), and the pH of the filtrate was

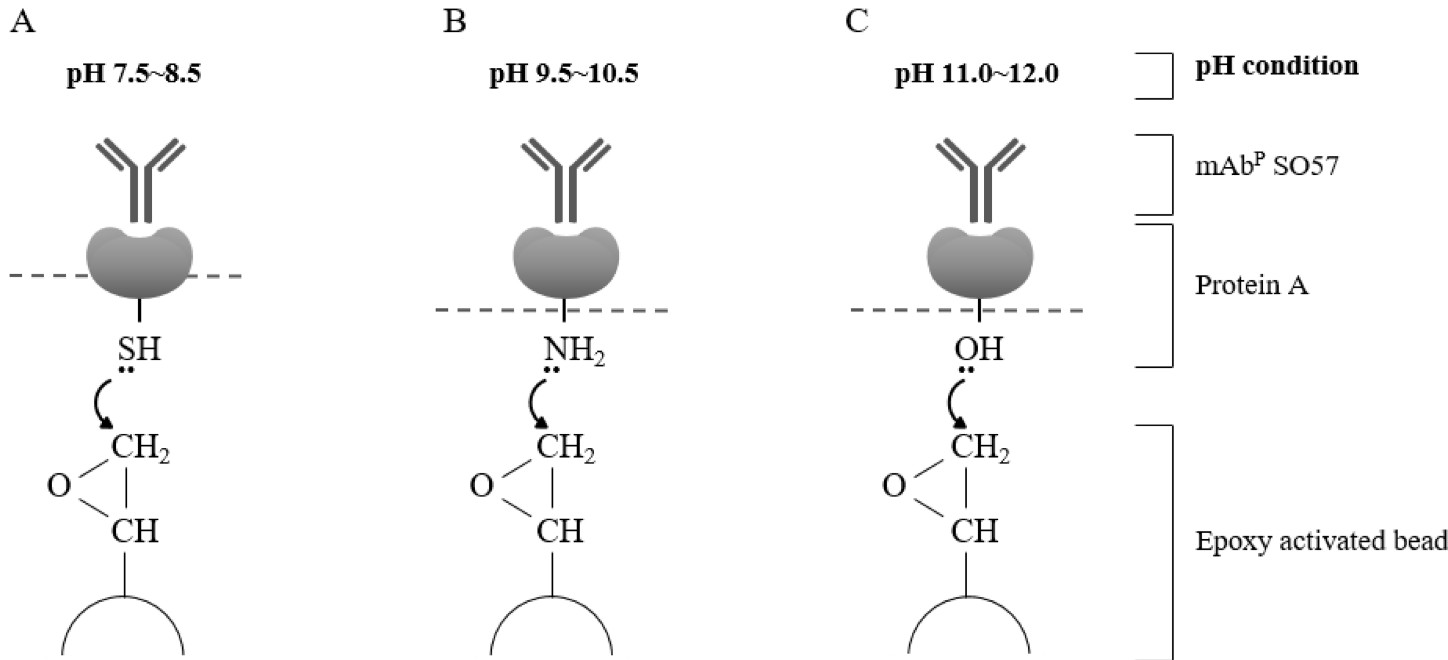

**Figure 3 Schematic diagram of pH-dependent covalent bonding of epoxy-activated agarose bead and different functional groups of recombinant protein A.** The epoxy group on agarose beads can react with sulfhydryl, amines, or hydroxyl groups of protein A according to pH conditions. Sulfhydryl groups are highly reactive with epoxide in the range of pH 7.5–8.5 (A), amine nucleophiles react at moderate alkaline pH (B), and hydroxyl groups require high pH conditions in the range of pH 11–12 (C).

reduced to 5.1 by adding acetic acid (pH 2.4). The solution was centrifuged to remove chlorophyll at $10,200 \times g$ for 30 min at 4 °C. The pH of the solution was then increased to 7.0 by adding 3M Tris-HCl, and 15% ammonium sulfate was added to the solution at 4 °C. After centrifugation at $9,000 \times g$ for 30 min at 4 °C, 45% of ammonium sulfate was added to the collected supernatant at 4 °C. After overnight incubation, the total solution was centrifuged at $10,200 \times g$ for 30 min at 4 °C. The pellet was resuspended in an extraction buffer with 1/12 volume of the original extraction buffer, and centrifuged at $10,200 \times g$ for 30 min at 4 °C. Final solutions were divided into five groups, and each solution was applied to four different resins and a control GE protein A resin for purification of mAb[P]SO57 from total soluble protein (TSP) solution (Fig. 2). Four resins were manufactured by using Sepharose 4 Fast Flow (GE Healthcare, Uppsala, Sweden). The Sepharose resin was washed on a filter funnel with distilled water and activated with 1,4-butanediol diglycidyl ether in 0.6M NaOH solution for 10 h at 25 °C. After washing, the activated resin was coupled with purified recombinant protein A (10 mg/mL resin) in a coupling buffer (100 mM Na-phosphate, 1 mM EDTA, pH 8.5, pH 9.5, pH 10.5, and pH 11.5) for 18 h at 37 °C. Each pH was adjusted with NaOH (Fig. 3). Lastly, these four resins were washed with distilled water several times and then deactivated with 1M ethanolamine pH 8.0.

## Protein concentration analysis

Eluted fraction F1–F7 of purified mAb[P]SO57 were quantified using Epoch spectrometer (UV–Vis Microplate Spectrophotometer; Biotech, Winooski, VT, USA) (*Oey et al., 2009*; *Song et al., 2015*). Aliquots of two μL samples were pipetted directly onto the pedestal

and measured at 206–280 nm. The concentrations of purified mAb$^P$SO57 at eluted fraction F1–F7 were calculated from the absorbance values by Gen5 2.01 software (Biotek, Highland Park, VT, USA).

### In vitro rabies virus neutralization assay

To compare neutralizing activities of purified mAb$^P$s obtained from different resins (8.5R, 9.5R, and 10.5R), rapid fluorescent focus inhibition test (RFFIT) was performed plant-derived mAb$^P$s (8.5R, 9.5R, and 10.5R) and anti-rabies immunoglobulin (National Institute for Biological Standards and Control, Potters Bar, UK), respectively. The mAbs, at an initial concentration of 20 μg/mL, were diluted in 1:2.5, 1:12.5, 1:62.5, and 1:312.5 with advanced Dulbecco's modified Eagle's medium (Sigma, St. Louis, MO, USA). A 100 μL aliquot of each diluted sample and 100 μL of Challenge Virus Standard (CVS-11) containing 32–100 FFD$_{50}$ were mixed in an 8-well chamber slide (Nunc, Rochester, NY, USA), performed in duplicate. These mixed samples were incubated in a 5% CO$_2$ incubator at 35 °C for 90 min. After incubation, $0.5 \times 10^5$ cells of mouse neuroblastoma (N2a) cells with 100 μL total volume were added to each chamber and incubated at 35 °C in a 5% CO$_2$ incubator for 20 h. After incubation, the supernatant was removed, and monolayers were fixed in 80% acetone for 10 min at 23 °C after washing with 1× PBS. The slides were stained with 0.0025% Evans Blue solution (ScienceLab, Houston, TX, USA), containing anti-rabies virus mAbs conjugated to Alexa Fluor$^{TM}$ 488 (Invitrogen, Carlsbad, CA, USA), and incubated at 37 °C for 30 min. After incubation, the slides were washed three times for 1 min with 1× PBS and then coverslips were mounted on each chamber slide. The slides were observed under a fluorescence microscope (Carl Zeiss, Oberkochen, Germany) (200× magnification).

## RESULTS

### Generation of Arabidopsis transformants expressing anti-rabies virus mAb$^P$SO57

A total of 2,000 T$_1$ seeds obtained after floral dip transformation were plated on MS medium containing kanamycin (50 mg/L) for selection of transformants. Of these seeds, 40 plants had true leaves with green color, whereas almost all the sown seeds failed to develop true leaves and were etiolated with light yellow shoots. Forty transformants with true leaves survived from kanamycin media, and were transplanted to a growth chamber in standard conditions (Fig. 1B). The rosette leaves of the 40 plants were used as material for the confirmation of transgene insertion. The expected PCR HC and LC gene bands were observed in all T$_1$ Arabidopsis plants (Fig. 1B). No PCR band was observed in wild type Col-0 Arabidopsis (data not shown). Repeated kanamycin selections were conducted in successive generations to find homozygous seeds for mass production of transgenic Arabidopsis plants expressing mAb$^P$SO57 (Fig. 1C).

### Expression of heavy chain and light chain proteins in transgenic Arabidopsis

After confirmation of transgene insertion, expression of HC and LC proteins in leaves was investigated by western blot analysis (Fig. 1B). The HC and LC protein bands were

detected approximately at 50 and 25 kDa, respectively. Among the 40 plants, 26 plants showed HC and LC protein bands (data not shown), and four plants (a-2, b-2, b-3, and b-4) showed high protein expression levels (Fig. 1B). $T_2$ seeds from the b-2 plant with high protein expression levels were obtained for further study.

## Purification of human anti-rabies mAb from Arabidopsis leaf tissue under different pH conditions

Different agarose resins coupled to protein A under pH 8.5, 9.5, 10.5, and 11.5 conditions were used to purify the anti-rabies mAbs from plant leaf biomass. Two steps were used, which were ammonium sulfate-mediated TSP precipitation, followed by protein A affinity chromatography (Fig. 2). These steps efficiently isolated the majority of TSP including rubisco protein and separated specific mAbs from plant leaf extracts. SDS–PAGE analysis was used to identify the HC (50 kDa) and LC (25 kDa) of the anti-rabies virus mAb[P]SO57 from elution fractions from plant TSP (Fig. 4). mAb[P]SO57 were eluted mainly in Fractions 1 and 2 in GE resin (GR), but intensively eluted in Fraction 4 in other resins (pH 8.5 (8.5R), 9.5 (9.5R), 10.5 (10.5R), and 11.5 (11.5R)) (Fig. 4).

## Quantification of purified mAb[P]SO57

Each eluted sample (F1–F7) was quantified using a protein concentration analysis (UV–Vis Microplate Spectrophotometer; BioTek, Winooski, VT, USA). The highest concentration of mAb[P]SO57 among seven fractions (F1–F7) of each experimental group were as follows: GR (F2, 310.5 μg/mL), 8.5R (F4, 485 μg/mL), 9.5R (F4, 550 μg/mL), 10.5R (F4, 410 μg/mL), and 11.5R (F4, 15 μg/mL), respectively (Fig. 5). The total amount of obtained mAbs were 350 μg (GR), 400 μg (8.5R), 360 μg (9.5R), 380 μg (10.5R), and 25 μg (11.5R), respectively (data not shown). 8.5R resins had the highest amount of purified mAb[P]SO57 in the other fractions compared to the other groups.

## Neutralizing activity of the purified mAb[P]SO57

The rabies virus-neutralizing activities of mAb[P]SO57 purified from different resins were compared to commercial antibodies (Anti-rabies human immunoglobulin). The mean RFFIT values of each group were as follows: GR (7.16 IU/mL), 8.5R (5.50 IU/mL), 9.5R (7.29 IU/mL), and 10.5R (8.29 IU/mL), respectively. The mAb[P] could not be purified from 11.5R resin (Fig. 6). Thus, 11.5R was omitted for this neutralization assay.

## DISCUSSION

Protein affinity purification was widely used to purify biopharmaceutical proteins based on specific surface interactions, efficiently separating biological molecules with target antibodies and immobilized protein ligands (*Kubota et al., 2017*; *Saraswat et al., 2013*; *Sheng & Kong, 2012*). In affinity chromatography, protein A as a ligand covalently bonded to an agarose matrix is used to separate a target protein from a protein pool including cell lysate. The agarose matrix should be activated for covalent binding of ligands such as protein A for affinity purification. The agarose resin is epoxide-activated by immobilization of oxiranes such as 1,4-butanediol diglycidyl ether onto the matrix (*Oliveira, Tamashiro & Bueno, 2015*; *Zucca, Fernandez-Lafuente & Sanjust, 2016*).

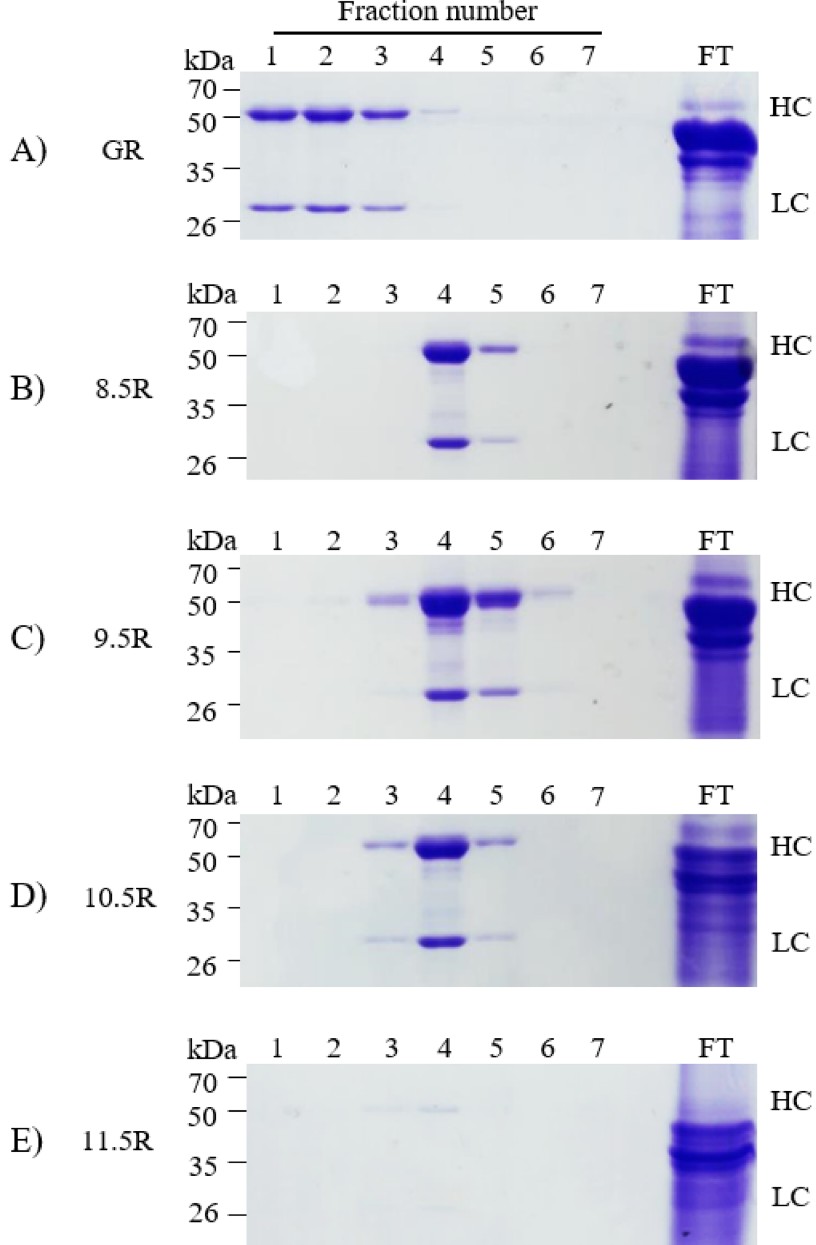

**Figure 4** **SDS–PAGE analysis of eluted F1–F7 fractions of purified samples obtained from transgenic *Arabidopsis* expressing mAb$^P$SO57 in reducing condition.** Lane 1–7, eluted fractions of the purification F1–F7, respectively; Lane 9, flow through; HC, heavy chain of mAb$^P$; LC, light chain of mAb$^P$. Epoxy-activated agarose beads were coupled to protein A under the pH conditions of 8.5, 9.5, 10.5, and 11.5 (8.5R, 9.5R, 10.5R, and 11.5R, respectively). Commercial protein A resin (GR) (GE Healthcare, Uppsala, Sweden) was used as a positive control.

The epoxide groups can be coupled with thiols, primary amines, and hydroxyl HOOC– groups depending on pH 8.5, 9.5, 10.5, and 11.5 conditions, respectively. The immobilized ligand such as protein A on an epoxy-activated agarose matrix can bind to mAbs (*Mazzer et al., 2015*; *Minakuchi et al., 2013*). The directionality of Protein A attached to resin is different depending on the immobilized residue. When protein A is

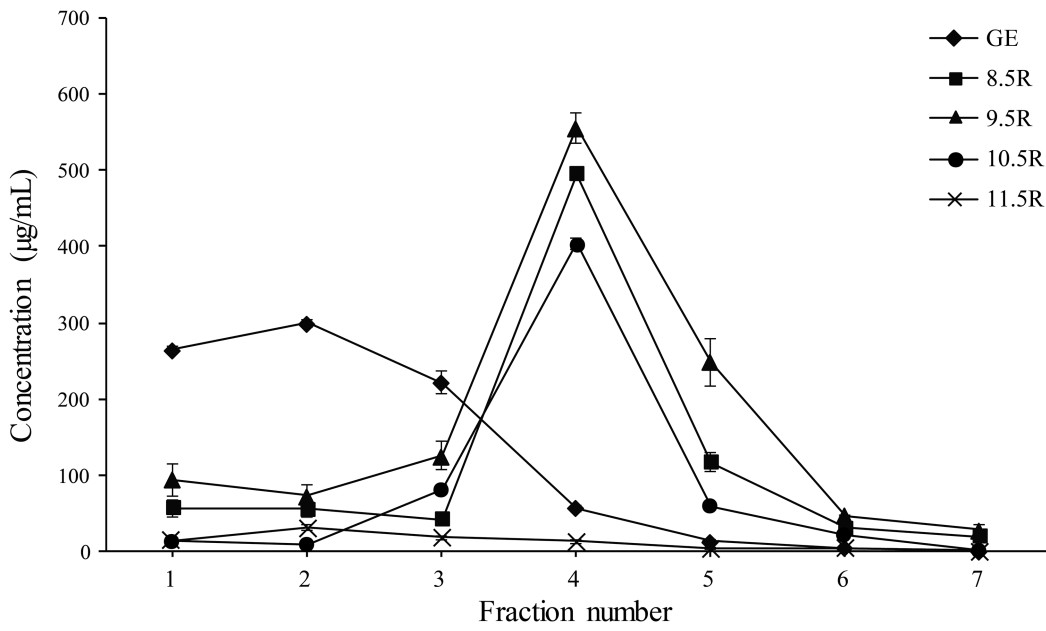

**Figure 5 Quantification analysis of eluted fraction F1–F7 of purified mAb$^P$SO57 using Epoch spectrometer.** The vertical axis values (μg/mL) represent the mean value of four-time measurement per each case. Diamond, positive control resin (GE) (GE Healthcare, Uppsala, Sweden); square, resins under pH 8.5 condition (8.5R); triangle, resins under pH 9.5 condition (9.5R); circle, resins under pH 10.5 condition (10.5R), and cross, resins under pH 11.5 condition(11.5R), respectively.

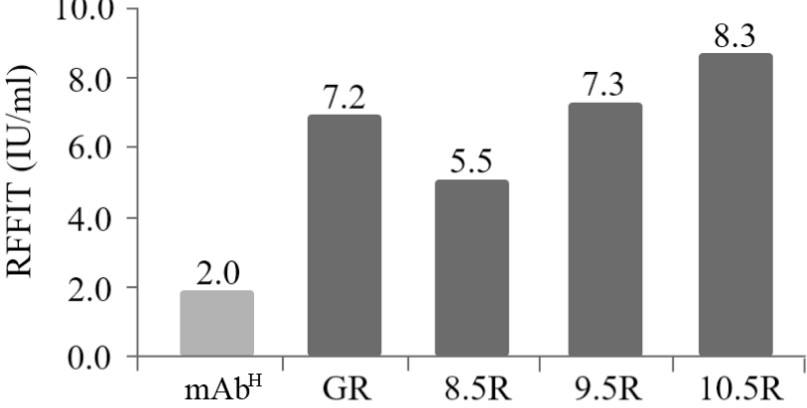

**Figure 6 Comparison of virus-neutralizing activity of mAb$^P$SO57 purified from resins differently coupled to protein A under pH 8.5, 9.5, 10.5, and 11.5 conditions (8.5R, 9.5R, 10.5R, and 11.5R, respectively) against target rabies viruses.** The values (IU/mL) represent the mean value of duplicate measurements. Commercial protein A resin (GR) (GE Healthcare, Uppsala, Sweden) was used as a positive control.

immobilized on the resin with amine ($NH_2^-$) and hydroxyl ($OH^-$) groups, the outward directionality of Protein A is random. Thus, we optimized the cysteine residue at the C terminal of protein A to give a constant directionality. When the protein A binds to the sulfhydryl group ($SH^-$), it is directed outward in a row. It is expected that the constant directionality can deliberately induce protein A to become saturated with the

resin by using a large amount of protein A and reacting for a long time. In this research, we manufactured four differently coupled the optimized protein A and epoxy-activated agarose resins to purify plant derived anti-rabies mAbs under different pH conditions. SDS–PAGE results showed that the capacity of pH 11.5 group was too low to effectively purify mAb[P]SO57 in transgenic *Arabidopsis* compared to that in the other experimental groups (pH 8.5, 9.5, and 10.5) including protein A Sepharose[TM] resin (GE Healthcare, Uppsala, Sweden). *Groher & Suess (2016)* speculated that the binding power of the hydroxyl groups and epoxy-activated agarose beads in pH 11.5 conditions is weakened by the hindrance of hydroxyl groups on the surface of the agarose beads. Epoxy-activated agarose coupled with protein A resin under pH conditions greater than 13 was not recommended for purification of mAbs (*Sada, 1990*). SDS–PAGE results showed that the four different pH conditions had longer retention time (intensively eluted from Fraction 4) than GE resin (intensively eluted from Fraction 1). We hypothesize that these retention time differences are due to epoxy-activated beads' structure and bead porosity. Indeed, the pore sizes of the four differently coupled resins with protein A were similar to that of the GE resin. It is speculated that under pH 8.5 conditions, the lower level of sulfhydryl residue of protein A compared to the amine residue induced holding the protein A in a specific orientation on the resin rather than in random directions (*Zhang, Duan & Zeng, 2017*), consequently enhancing the mAb binding capacity (*Batalla et al., 2012*; *Liu & Yu, 2016*). However, SDS–PAGE results showed that the effect of pH conditions on coupling with epoxy beads did not affect the purity of plant-derived mAb[P]SO57. Although the 8.5R, 9.5R, and 10.5R showed some non-specific bands below 50 kDa, the purified quantity are more than two times stronger than the GE. The virus-neutralizing efficacies of four mAb[P]SO57 purified from different resins (GE, pH 8.5, pH 9.5, pH 10.5) were approximately three times higher than that of anti-rabies human immunoglobulin. Among the purified plant-derived antibodies samples, purified sample with the 10.5 resin showed the highest RFFIT value, indicating that the 10.5 resin purified more active mAb form compared to the others.

## CONCLUSION

In this study, the effect of pH on coupling between protein A and epoxy-activated beads was investigated with transgenic *Arabidopsis* expressing mAb[P]SO57. Overall, our results suggest that both pH 8.5 and 10.5 are optimal for the purification of anti-rabies mAbs from transgenic *Arabidopsis* plant leaf. This is the first research report on optimization of protein A chromatography resin using manipulation of coupling with pH conditions. Plant expression system has emerged as one of the most promising platforms for the large-scale production of therapeutic reagents (*Sharma & Sathishkumar, 2017*). Several major companies have occupied the most proportion for the commercial resin market (*Pollock et al., 2017*). In addition, the cost of commercialized affinity resin product is relatively high (*Buyel, Twyman & Fischer, 2017*). The currently described resin, in this study, was confirmed to have equivalent qualities compared to commercially available protein A resin. For a commercial use of plant as a bioreactor, resin cost which occupies a large portion of down-stream cost should be reduced in the future (*Arora, Saxena & Ayyar, 2017*). By using

our currently described cost effective resins, bulk production of therapeutic mAbs using plant system would be realized in the future.

### Funding

This research was supported by a grant (Code#PJ0134372019) from the Korean Rural Development Administration, by the Technology Innovation Program (10079457) funded by the Ministry of Trade, Industry & Energy (MI, Korea), and the Basic Science Research Program through NRF funded by the Ministry of Education (NRF-2016R1A6A3A11930180). The funders had no role in study design, data collection and analysis, decision to publish, or preparation of the manuscript.

### Grant Disclosures

The following grant information was disclosed by the authors:
Korean Rural Development Administration: Code#PJ0134372019.
Technology Innovation Program: 10079457.
Ministry of Trade, Industry & Energy (MI, Korea), and Basic Science Research Program through NRF.
Ministry of Education: NRF-2016R1A6A3A11930180.

### Competing Interests

Su-Lim Choi is employed by Amicogen, Korea. The other authors declare there are no competing interests.

### Author Contributions

- Ilchan Song conceived and designed the experiments, analyzed the data, contributed reagents/materials/analysis tools, prepared figures and/or tables, authored or reviewed drafts of the paper, approved the final draft.
- Yang Joo Kang performed the experiments, analyzed the data, contributed reagents/materials/analysis tools, prepared figures and/or tables, authored or reviewed drafts of the paper.
- Su-Lim Choi conceived and designed the experiments, analyzed the data, contributed reagents/materials/analysis tools, prepared figures and/or tables, approved the final draft.
- Dalmuri Han performed the experiments, analyzed the data, contributed reagents/materials/analysis tools, prepared figures and/or tables.
- Deuk-Su Kim performed the experiments, analyzed the data, contributed reagents/materials/analysis tools, prepared figures and/or tables.
- Hae Kyung Lee performed the experiments, analyzed the data, contributed reagents/materials/analysis tools, prepared figures and/or tables, approved the final draft.
- Joon-Chul Lee conceived and designed the experiments, analyzed the data, contributed reagents/materials/analysis tools, authored or reviewed drafts of the paper, approved the final draft.
- Jeanho Park conceived and designed the experiments, analyzed the data, contributed reagents/materials/analysis tools, authored or reviewed drafts of the paper, approved the final draft.
- Do-Sun Kim authored or reviewed drafts of the paper, approved the final draft.
- Kisung Ko conceived and designed the experiments, performed the experiments, analyzed the data, contributed reagents/materials/analysis tools, prepared figures and/or tables, authored or reviewed drafts of the paper, approved the final draft.

## Data Availability

Raw data are provided in the Supplemental Materials.

## Supplemental Information

Supplemental information for this article can be found online at http://dx.doi.org/10.7717/peerj.6828#supplemental-information.

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
