# Peer review of "Purification of plant-derived anti-virus mAb through optimized pH conditions for coupling between protein A and epoxy-activated beads"

_PeerJ, doi:10.7717/peerj.6828_

## Round 0.1 · original submission · Major Revisions

Please make sure to respond to the specific points raised by the reviewers.

·

Basic reporting

No comment

Experimental design

1. Optimize the time and temperature for the proteinA couple resin, Why choosing 18 h at 37°C ?
2. The antibody should be quantify by ELISA, not measure total protein. Because you need to compare using the same amount of the full assembly antibody

Validity of the findings

1.The result showed clear conclusion that using pH8.5 to conjugate the proteinA to the bead will give good result in antibody purification. However, the authors should repeat the experiment to be able to do the statistical analysis. This will improve the conclusion. The significant finding of this study is to confirm that which condition is the best among all condition. Therefore, the statistical analysis is needed.
2.Provide the cost of protein A, to claim that the described resin will be cheaper than the commercial protein A resin.
3. In line 262, The virus-neutralizing efficacies of four mAbPSO57 purified from different resins (GE, pH 8.5, pH 9.5, pH 10.5) are similar and approximately three times higher than that of anti-rabies human immunoglobulin.

What is the source of the anti-rabies human immunoglobulin using in this study. How to make sure that the amount of the antibody is equal to the antibody purified from different pH resin? To conclude that the antibody purified from different pH resin is higher than the anti-rabies human immunoglobulin, the authors have to use the same amount of the antibody.

Additional comments

1. In line 250 the protein A with the different pH coupling conditions might hold both IgG Fc and IgG F(ab′)2, compared to the GE protein A which holds only Fc …What is the evidence for this claim?
2. In line 250, It is speculated that the protein A with the different pH coupling conditions might hold both IgG Fc and IgG F(ab′)2, compared to the GE protein A which holds only Fc. Why proteinA with different pH can hold Fab? Any evidence?
3. In line 257, The fusion of the ER retrieval modif, KDEL, to protein allow for their efficient retention in the ER. This is not related with the investigation at all. Because the authors used only one construct that contain KDEL.

Reviewer 2 ·

Basic reporting

Article is well written, and is a stand-alone study on the relevance of pH on the coupling of Protein A to agarose resin for the purpose of chromatographic separation of mAbs. There are some small errors, but overall the article is written well.

Experimental design

The research question is defined, however, the pH of the running buffer was not well indicated. It appears as though the running buffer for chromatography was the same pH as that of the coupling buffer. The method indicates that the precipitated pellet was reconstituted into “an extraction buffer” (lines 137 and 138). Is this the same buffer at pH 7.5? If the pHs were different, I don’t see how the authors can draw any of their conclusions from the work. If they were all at the same pH, then that would be slightly different. Each run seems to have been performed multiple times, though the statistics on the in vitro assay are unclear.

Validity of the findings

- The impact of the research appears to be a better way to couple Protein A to agarose resin. However, there is no assessment of the coupling efficiency that I can see, meaning that the conclusion of better placement of the Protein A ligands may be called into question.
- The article states that the better pH for coupling is at 8.5, but ignores the fact that the 10.5 resin had a higher RFFIT value. Meaning, that though efficiency of total protein may be better for the 8.5 resin, the 10.5 resin appears to have a higher activity.
- There was no densitometry done on the gels to indicate a purity level. All calculations were done using absorbance, which is a measure of total protein (and DNA in many cases) only. There appears to be no purity advantage to the resin made at pH 8.5
- The in vitro assay performed does not indicate whether the fractions are pooled or not. If they are, then this would be a clear case to prefer the 10.5 pH resin.

Data is robust, conclusions (excepting the above) are sound.

Additional comments

The article is a well-written account of the effects of varying pH levels on the recovery of the mAb expressed in arabidopsis plants. There may need to be more characterization of the eluates before some of the conclusions can be drawn. I currently see no advantage to the pH 8.5 resin vs the 10.5. I would also appreciate a comparison of ligand density between the test resins and the positive control to determine if that might be the reason for increased efficiency.

Reviewer 3 ·

Basic reporting

The quality of written English is good.

The article is appropriately structured and the figures are clearly reproduced. Raw data is provided.

There are a large number of references, perhaps too many. Some references are misformatted in the bibliography (e.g. Sharma and Sathishkumar – no journal, various journal names incorrectly capitalised, Hermanson GT 2013 – incomplete reference, no publisher)

Other minor issues:
L118 … based 5% skim milk. What does ‘based’ mean here?
L120 Supplier and product number of 2ndary antibody not stated
L150, 209, Figure 5 Nanodrop is a trademark of Thermo Scientific and should not be used to describe other spectrophotometers
L160 ADMEM supplier?
L183 ‘Repeating’ should read repeated, or sentence rephrased.
L244 ‘Epoxy-activated agarose coupled with protein A resin under pH conditions greater than 11 was not recommended for purification of monoclonal antibodies.’ Please provide reference.

Experimental design

This is a straightforward method development study comparing the ability of affinity matrices prepared under different conditions to bind mAb produced in transgenic Arabidopsis extracts. Samples are appropriate to allow comparisons between the yield and quality of antibodies eluted from different resins. The RFFIT test is an appropriate measure of functionality given the antibody’s specificity.

Validity of the findings

There are some weaknesses in the results presented for the antibody assays:

In Figure 4, it is not stated if these gels were run under reducing conditions (as they appear to be). Given the authors speculation that different protein A binding chemistries may predispose the resin to purifying F(ab’)2 fragments (L250) and also to better assess the purity intact mAb, non-reducing gel images should also be provided.

In Figure 5, error bars should be shown to illustrate the precision of the measurements. Is there a significant difference between yields under 10.5R, 9.5R and 8.5R conditions? Some calculations are made in the supplemental spreadsheet but only 2 measurements are provided instead of triplicates as stated in the Figure legend?

In Figure 6 again there are no error bars. Can the author say if there is any real difference between these samples on the RFFIT test? An appropriate statistical test should be performed.

L241 I feel the Groher & Suess 2016 reference is overinterpreted, the authors of that study simply say that binding via hydroxyl groups requires a highly basic pH which ‘not all ligands can withstand’ and do not mention protein A.

L256 -L261 the authors discuss the influence of including signal peptides and KDEL motifs on the antibody sequence but no connection is made to the results presented here, or even to protein A binding. This section should be removed.

Additional comments

Although manufacturers often provide instructions for performing immobilisations similar to that performed here to generate protein A affinity matrices, there is little practical information in the scientific literature to support these general protocols.

I believe this MS will be of practical use to a number of laboratories aiming to produce these columns as a tool for their research, and is suitable for publication if weaknesses in the assay data can be addressed. Addressing these concerns is important, as the data as currently presented does not fully support the conclusion that pH 8.5 is optimal.

Although the authors place importance on the production platform for the mAb (Arabidopis plants), I suspect these results will be applicable to other production methodologies. Perhaps the authors would like to speculate on this point?

---

## Round 0.2 · Minor Revisions

One of the reviewers has raised an important point for the clarity of the text and merit of the work, related to the aa composition. Please revise accordingly.

·

Basic reporting

The manuscript is clear with sufficient context provided.

Experimental design

The revised manuscript has good explanation on the experimental design.

Validity of the findings

The study showed the relevance of pH on the coupling of Protein A to agarose resin for the purpose of chromatographic separation of mAbs.

Additional comments

I think the revised manuscript is good to publish.

Reviewer 3 ·

Basic reporting

Improvements have been made to raw data and the references as requested in revision. Other aspects to an acceptable standard, as before.

The minor missing details, etc, in my comments have been corrected.

Experimental design

Appropriate as before

Validity of the findings

Key improvements and clarifications have been made to Figures 4 and 5 and the discussion. These have largely addressed my concerns in my first review.

However, it struck me in a response the authors made to reviewer 2, that the recombinant protein A used in these experiments had in fact been engineered to contain a cysteine residue at the C terminus:

'...Therefore, we authors has optimized the cysteine residue at the C terminal of protein A to give a constant directionality. (Original protein A has no cysteine residue).'

This was not at all obvious from the original MS and clearly has implications for groups who may wish to produce equally effective columns with unmodified protein A. Please clarify this point in the MS and specify the modification(s) made.

---

## Round 0.3 · accepted · Accept

I am one of the Section Editors, and I have taken over this submission as the Editor is unavailable to make a decision.

All the critical points raised by the reviewers were adequately addressed and the manuscript was appropriately amended. The revised version is acceptable.

#